# Planned Diffusion

**Daniel Israel**[1][*]   **Tian Jin**[2][*]   **Ellie Cheng**[2]   **Guy Van den Broeck**[1]
**Aditya Grover**[1]   **Suvinay Subramanian**[3]   **Michael Carbin**[2]

[1]University of California, Los Angeles   [2]MIT CSAIL   [3]Google

## Abstract

Most large language models are autoregressive: they generate tokens one at a time. Discrete diffusion language models can generate multiple tokens in parallel, but sampling from them requires a *denoising order*: a strategy for deciding which tokens to decode at each step. Determining a good denoising order is difficult, and existing approaches use heuristics that create a steep trade-off between quality and latency. We propose *planned diffusion*, a system that trains the model to determine its own denoising order. Planned diffusion uses a single model that transitions between autoregressive and diffusion-based generation: first, the model autoregressively generates a plan that partitions the response into semantically independent chunks; second, the model denoises all chunks in parallel. The autoregressive plan enables the model to define the denoising order itself. On AlpacaEval, planned diffusion achieves 1.27x to 1.81x speedup over autoregressive generation with only 0.87% to 5.4% drop in win rate, establishing a new Pareto frontier for parallel generation with discrete diffusion. Additionally, planned diffusion's instruction-following quality continues to improve with more finetuning compute, while the autoregressive baseline plateaus. Our implementation provides simple runtime knobs that offer tunable control over the quality-latency trade-off.

## 1 Introduction

Large language models achieve strong performance on diverse tasks such as open-ended dialogue (Ouyang et al., 2022), code generation (Chen et al., 2021), and mathematical reasoning (Lewkowycz et al., 2022). Most existing large language models are autoregressive (Brown et al., 2020; Chowdhery et al., 2023; Touvron et al., 2023): they generate text one token at a time, and cannot decode any token until they have decoded every token before it. This sequential dependence between decoding steps within a response creates a fundamental latency bottleneck. Discrete diffusion language models (Sahoo et al., 2024; Shi et al., 2024) take a different approach: they decode multiple tokens per step. However, sampling from discrete diffusion models requires a *denoising order*: the strategy for deciding which tokens to decode at each step. Determining a good denoising order is difficult, and existing approaches use heuristics such as random order or confidence-based thresholds, creating a steep trade-off between quality and latency (Wu et al., 2025).

We propose *planned diffusion*, a system that trains the model to determine its own denoising order. Typical language model responses contain semantically independent chunks that the model can denoise in parallel. For example, in a response with a bulleted list, the model can denoise each bullet point concurrently (Ning et al., 2024). Planned diffusion exploits this by having the model identify independent chunks itself and denoise tokens in different chunks in parallel.

Planned diffusion uses a single model that transitions between autoregressive and diffusion-based generation. First, in an autoregressive planning stage, the model produces a generation plan composed of structural control tags. This plan partitions the response into a set of semantically independent chunks, defining a denoising order so that tokens in different chunks can denoise in parallel. Second, in a parallel diffusion stage, the model executes this denoising order, filling in all planned chunks simultaneously via diffusion denoising. This single-model, hybrid approach offers a distinct architectural advantage over other acceleration techniques, such as speculative decoding (Leviathan et al., 2023), which requires a separate draft model. To the best of our knowledge, we are the first to

---

[*]Equal contribution.

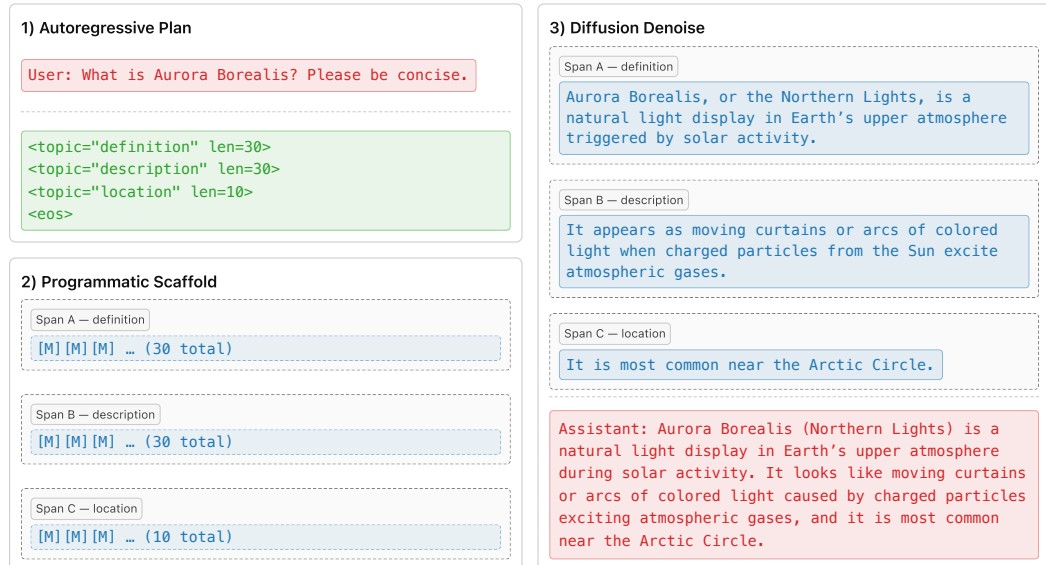

Figure 1: A real example of planned diffusion. (1) The model first generates a sequential plan using control tags to define the structure and length of independent text chunks. (2) The runtime then translates this plan into a scaffold, initializing each chunk with a corresponding number of mask tokens. (3) The model denoises all chunks in parallel with diffusion, generating the text for each section simultaneously to produce the complete response.

train a text-only model with both discrete diffusion and autoregressive objectives. Figure 1 presents a sample generation from our planned diffusion model.

We make the following contributions:

1. We introduce *planned diffusion*, a new parallel generation technique that decomposes text generation into a sequential planning stage and a parallel diffusion stage.
2. We design the control tags, training methodology, and inference algorithm that enable a single model to perform this hybrid generation process.
3. We demonstrate that planned diffusion achieves a Pareto-optimal trade-off between quality and latency on AlpacaEval, a suite of 805 instruction-following prompts; it achieves a 1.27x to 1.81x speedup over autoregressive generation while incurring only a 0.87% to 5.4% drop in win rate, respectively.
4. We show that planned diffusion's instruction-following quality continues to improve with more finetuning compute, while the autoregressive baseline plateaus.
5. We present sensitivity analysis validating that the planning mechanism is minimal and reliable, and that simple runtime knobs offer tunable control over the quality–latency trade-off.

## 2 RELATED WORK

Our work builds upon recent developments in diffusion-based language models and parallel text generation. We position our contributions in the context of three primary research areas: diffusion language models, semantic parallelism, and other parallel generation techniques.

**Diffusion Language Models.** Diffusion models have recently emerged as a new paradigm for generative language tasks (Austin et al., 2021; Sahoo et al., 2024; Lou et al., 2024; Shi et al., 2024; Nie et al., 2025; Ye et al., 2025; Liu et al., 2025a). A significant body of research is focused on accelerating the inference process, which traditionally involves many iterative denoising steps. These acceleration techniques include KV caching for diffusion models (Ma et al., 2025; Liu et al., 2025d), the use of autoregressive verification (Hu et al., 2025; Israel et al., 2025a), and the development of fast sampling strategies that reduce the number of required steps (Wu et al., 2025; Li et al., 2025; Hong et al., 2025). Planned diffusion complements these acceleration techniques and can integrate any diffusion sampling strategy, enabling further latency reductions. Other related works include block diffusion (Arriola et al., 2025), which enforces an autoregressive structure over blocks, and

planned denoising (Liu et al., 2025c), which learns an adaptive denoising schedule. While relevant, neither targets semantic parallelism, which is the purpose of planned diffusion.

**Semantic Parallelism.** Diffusion models parallelize token-level denoising but do not learn to exploit semantic independence across larger chunks of text. We define *semantic parallelism* as a broad class of techniques that produce models capable of parallelizing over semantically independent chunks of tokens.

Many recent works explore semantic parallelism (Ning et al., 2024; Liu et al., 2024; Jin et al., 2025; Rodionov et al., 2025; Pan et al., 2025; Wen et al., 2025; Yang et al., 2025). These works all rely on autoregressive decoding within each chunk; planned diffusion is, to the best of our knowledge, the first to instead denoise each chunk via diffusion (see Appendix E for further discussion).

**Other Parallel Generation Techniques.** Insertion-based models generate text by predicting where and what tokens to insert in a sequence. Multiple insertions can occur simultaneously, reducing the total number of decoding steps (Stern et al., 2019). Insertion-based models parallelize at the token level; planned diffusion instead learns to identify independent chunks and denoises them concurrently. Speculative decoding accelerates autoregressive models by drafting multiple tokens and verifying them in parallel (Leviathan et al., 2023; Chen et al., 2023; Zhang et al., 2024). Planned diffusion requires no separate draft model — a single model handles both autoregressive planning and parallel generation.

## 3  PRELIMINARIES

Generative language models learn a probability distribution over sequences of discrete tokens. In this work, we focus on two main paradigms: autoregressive and discrete diffusion.

**Autoregression.** Autoregressive models are the standard for sequential text generation. They factorize the joint distribution $p_{\text{AR}}$ of a token sequence $x = (x_1, x_2, \cdots, x_L)$ as a product of conditionals:

$$p_{\text{AR}}(x) = \prod_{i=1}^{|x|} p_\theta(x_i | x_{<i}) \tag{1}$$

where $p_\theta$ is a parameterized conditional distribution over tokens. In autoregression, the decoding algorithm sequentially samples each token conditioned on all previously generated tokens.

**Discrete Diffusion.**  Discrete diffusion models learn to reverse a predefined data corruption process that gradually introduces noise into a clean sequence. For text, this involves incrementally replacing tokens with a special *mask* token (Austin et al., 2021). Let $x$ be a clean sequence of $L$ tokens. The forward corruption process $q$ produces a noisy version $\tilde{x}$ at a noise level $t \in [0, 1]$, masking each position independently.

$$q_{t|0}(\tilde{x}_i \mid x_i) = \begin{cases} t, & \text{if } \tilde{x}_i = [\text{MASK}] \\ 1 - t, & \text{if } \tilde{x}_i = x_i \\ 0 & \text{otherwise} \end{cases} \qquad q_{t|0}(\tilde{x} \mid x) = \prod_i q_{t|0}(\tilde{x}_i \mid x_i). \tag{2}$$

Because the true posterior over the forward noising process is intractable (Lou et al., 2024), diffusion models learn an approximation by maximizing a variational lower bound on the log-likelihood (Sahoo et al., 2024; Shi et al., 2024).

$$\log p_{\text{D}}(x) \geq \mathbb{E}_{t \sim U(0,1), \, \tilde{x} \sim q_{t|0}(\tilde{x}|x)} \frac{1}{t} \sum_i \mathbb{1}(\tilde{x}_i = [\text{MASK}]) \log p_\theta(x_i \mid \tilde{x}). \tag{3}$$

Because the training objective decomposes over individual positions (Equation 3), for any disjoint sets of observed positions $\mathcal{O}$ and queried positions $\mathcal{Q} \subseteq [L]$, the conditional distribution over $\mathcal{Q}$ factorizes as

$$p_{\text{D}}(x_{\mathcal{Q}} \mid x_{\mathcal{O}}) = \prod_{i \in \mathcal{Q}} p_{\text{D}}(x_i \mid x_{\mathcal{O}}). \tag{4}$$

Planned diffusion conditions both distributions on prior context tokens $c = (c_1, c_2, \cdots)$; we denote such conditional distributions as $p_{\text{AR}}(\cdot \mid c)$ and $p_{\text{D}}(\cdot \mid c)$.

**Denoising Order.** Sampling from a discrete diffusion model requires selecting a *denoising order* (Kim et al., 2025; Turok et al., 2026). We define an *ordered partition* of positions $[L] = \{1, \ldots, L\}$ as a tuple $\sigma = (\sigma_1, \ldots, \sigma_T)$ of non-empty disjoint subsets with $\bigcup_{t=1}^{T} \sigma_t = [L]$. We write $\sigma_{<t} = \sigma_1 \cup \cdots \cup \sigma_{t-1}$ for positions revealed before step $t$. Positions within the same subset decode in *parallel*; positions in different subsets decode *sequentially*. A denoising order selects an ordered partition of $[L]$, determining how the joint distribution factors across decoding steps:

$$p_{\mathrm{D}}(x \mid c; \sigma) = \prod_{t=1}^{T} p_{\mathrm{D}}(x_{\sigma_t} \mid x_{\sigma_{<t}}, c). \tag{5}$$

obtained by substituting $\mathcal{O} = \sigma_{<t}$ and $\mathcal{Q} = \sigma_t$ into Equation 4. We define *uniform* and *entropy* based ordering. For simplicity, we only define the orderings where $|\sigma_t| = 1$ for all $t$. Let $\pi$ be a permutation of $[L]$ drawn uniformly at random. The uniform ordered partition is

$$\sigma_t^{\mathrm{Unif}} := \{\pi(t)\} \qquad t = 1, \ldots, L \tag{6}$$

Uniform random unmasking is the denoising order implied by the training objective in Equation 3, but in practice it underperforms entropy-ordered unmasking (Israel et al., 2025a). The entropy-ordered partition selects at each step $t$ the position with lowest predicted entropy

$$\sigma_t^{\mathrm{Ent}} := \underset{i \notin \sigma_{<t}}{\arg\min} \, H\big(p_{\mathrm{D}}(x_i \mid x_{\sigma_{<t}})\big) \qquad t = 1, \ldots, L \tag{7}$$

where $H$ denotes Shannon entropy. Unlike $\sigma^{\mathrm{Unif}}$, the partition is not fixed in advance but depends on the tokens revealed at each step. This is the default inference strategy of Ye et al. (2025).

## 4 PLANNED DIFFUSION

Planned diffusion alternates between autoregressive planning and parallel diffusion denoising. Each *iteration* consists of a single planning stage followed by a single diffusion stage; later iterations may condition on the outputs of earlier ones. The plan segments the output into chunks and concisely describes each chunk's semantic content; the diffusion model then fills in the chunks in parallel. Within each chunk, planned diffusion decodes following a separate inner denoising order, composing the parallel order over chunks with the inner order within each chunk.

### 4.1 FORMAL DESCRIPTION

We formalize a single iteration of planned diffusion. Beginning from prior tokens $c$, the model autoregressively generates a *plan* $z$, which specifies $K$ chunks, each with a semantic description, and predicted lengths $l_1, \ldots, l_K$. This induces a total sequence length $L = \sum_{k=1}^{K} l_k$ and a partition of $[L]$ into sets of contiguous indices $s_1, \ldots, s_K$, where each $s_k = \{L_{k-1}+1, \ldots, L_k\}$ is the set of indices starting from the end of the previous set until index $L_k = \sum_{j=1}^{k} l_j$ with $L_0 = 0$. We define an inner denoising order for each chunk by letting $\sigma^{(k)} = (\sigma_1^{(k)}, \ldots, \sigma_{T_k}^{(k)})$ be any ordered partition of $s_k$ into $T_k \leq l_k$ steps. Given this, the composed order $\sigma^{\mathrm{Plan}}(z)$ determines a joint distribution over plan $z$ and tokens $x$ conditioned on previous tokens $c$.

$$\sigma_t^{\mathrm{Plan}}(z) = \bigcup_{\{k \, : \, T_k \geq t\}} \sigma_t^{(k)}, \qquad t = 1, \ldots, \max_k T_k \tag{8}$$

which reveals positions from every chunk $s_k$ with $T_k \geq t$ simultaneously at step $t$. The diffusion phase generates content $x \in \mathcal{V}^L$, and the joint distribution over plan and content is:

$$p_{\mathrm{PD}}\big(z, x \mid c; \sigma^{\mathrm{Plan}}(z)\big) = \underbrace{p_{\mathrm{AR}}(z \mid c)}_{\text{Planning}} \cdot \underbrace{p_{\mathrm{D}}\big(x \mid z, c; \sigma^{\mathrm{Plan}}(z)\big)}_{\text{Diffusion}} \tag{9}$$

Because $p_{\mathrm{D}}$ accepts any valid ordered partition of $[L]$, planned diffusion decouples the choice of inner order $\sigma^{(k)}$ from the outer order over chunks. Setting each $\sigma^{(k)} = \sigma^{\mathrm{Ent}}$ restricted to $s_k$ recovers the entropy strategy applied independently within each chunk, which we use in our experiments. Algorithm 1 presents the full planned diffusion sampling algorithm, which can perform multiple iterations of planning and diffusion.

---

**Algorithm 1** Planned Diffusion

---

1: **function** PLANNED_DIFFUSION($c$)
2:     **loop**
3:         Sample plan $z \sim p_{\mathrm{AR}}(\cdot \mid c)$                            ▷ Autoregressively generate plan
4:         Parse $z$ to get $K$ chunks $s_1, \ldots, s_K$ with lengths $l_1, \ldots, l_K$
5:         **for** $k = 1, \ldots, K$ **do in parallel**
6:             Sample $x_{s_k} \sim p_{\mathrm{D}}\left(\cdot \mid z, c; \sigma^{(k)}\right)$             ▷ Diffuse each chunk in parallel
7:         **end for**
8:         $x \leftarrow (x_{s_1}, \ldots, x_{s_K})$                   ▷ Concatenate chunks together
9:         $c \leftarrow (c, z, x)$                ▷ Append plan and output to prior tokens
10:        **if** $z[\text{end}] = $ `<eos/>` **then**
11:           **break**               ▷ Final planning token triggers end of generation
12:        **end if**
13:     **end loop**
14:     $c \leftarrow$ REMOVECONTROLTAGS($c$)         ▷ Strip planning tokens from final output
15:     **return** $c$
16: **end function**

---

## 4.2 IMPLEMENTATION

**Control Tags.** Chunks denoting groups of semantically related tokens are first defined during a sequential planning stage. This is done using a paired `<topic>...</topic>` tag structure. Within this structure, the model generates a concise description of the chunk's content (e.g., "definition" in Figure 1) and its predicted length (e.g., "30" in Figure 1). During the parallel diffusion stage, the model generates the tokens for each chunk within a corresponding `<async>...</async>` tag pair. Finally, the `<sync/>` tag conveys generation dependency: tokens that follow `<sync/>` may require details produced inside the preceding `<async>` chunks, so the sampling algorithm continues sequential planning only after those chunks are filled and available. We add all control tags to the model's vocabulary for training and inference and strip them during post-processing of final outputs.

**Finetuning Dataset.** We annotate the SlimOrca instruction–finetuning dataset (Lian et al., 2023) for parallel generation, following Jin et al. (2025). We prompt a GEMINI model with the syntax and semantics of the control tags. See Appendix A for more details. GEMINI inserts the control tags into the response from each instruction-response pair in the dataset. The opening `<async>` carries two attributes: `topic` (a concise label, $\leq 3$ words) and `tokens` (a coarse length estimate, e.g., multiples of 10 tokens). We impose that every non-control-tag token lies inside exactly one `<async>...</async>` chunk (no nesting, no overlap). During preprocessing for training, we insert 0–10 tokens of stochastic padding inside each `<async>` chunk, allowing the diffusion model to generate with variable length shorter than the masked input. We validate well-formedness (balanced tags, coverage, non-overlap, attribute types/ranges) and discard malformed cases. Figure 5 contains a concrete example of the tagging language used by planned diffusion.

**Training Loss.** The goal of training is to maximize the joint probability over plan tokens and their content. Given an annotated dataset $\mathcal{D}$ in which a single clean example is given by $Y \in \mathcal{D}$, we decompose $Y$ into sets of planning tokens $Z$ and content tokens $X$, such that $Y = Z \cup X$. $X^t$ is a noised sequence of tokens under noise distribution $q_{t|0}$ as previously defined. Thus, $X^t$ contains masked tokens with probability $t$. Let $f_\theta$ be the function to instantiate planned diffusion, parameterized by $\theta$. We use the notation $f_\theta(x, i)$ to signify that the model takes input $x$ and makes a prediction at index $i$ of the sequence. Finally, let $M_i(X)$ be an attention masking function that takes as input a sequence $X$ and outputs a subset $M_i(X) \subseteq X$ of the sequence that $f_\theta$ has access to at a particular index $i$. We describe the attention masking in more detail in the following paragraph. With CE as the cross-entropy loss, our overall training objective is given by

$$\mathcal{L}(\theta) = \mathop{\mathbb{E}}_{\substack{Y \sim \mathcal{D} \\ t \sim U(0,1)}} \frac{1}{|Y|} \sum_{y_i \in Y} \underbrace{\mathbb{1}(y_i \in Z) \mathrm{CE}(f_\theta(y_{<i}, i), y_i)}_{\text{Autoregressive}} + \underbrace{\frac{1}{t} \mathbb{1}(y_i \in X) \mathrm{CE}\big(f_\theta(M_i(X^t \cup Z), i), y_i\big)}_{\text{Diffusion}}$$

$$(10)$$

Note that in the training objective, the same noise parameterized by $t$ is applied to chunks across multiple iterations of planning and diffusion, where future diffusion chunks will be conditioned on previously sampled diffusion chunks. In this decision, we are applying the interpretation of a

diffusion model as an any-order autoregressive model capable of supporting arbitrary conditional queries at inference time (Shi et al., 2024). The same technique is used by Llada 7B (Nie et al., 2025), which is trained with a diffusion objective, but during inference used for semi-autoregressive block sampling.

**Attention Mask.** We implement the following rules via the attention mask $M_i$. First, planning tokens, which are composed of control tags and their attributes, are given causal attention (Plan 1 and 2 in Figure 5). Second, diffused tokens use bidirectional attention, as required for diffusion-based parallel denoising (Diffusion 1 and 2 in Figure 5). In Section 5, we also show that restricting bidirectional dense attention within each chunk is another way to train planned diffusion. We refer to this variant as *planned diffusion sparse attention* (PSDA), which enforces full independence between chunks. The attention mask for PDSA can be seen in Appendix H.

**Variable Length Denoising.** Typically, diffusion models are configured to generate given a fixed number of denoising steps. Generation quality increases and speed decreases with the number of steps. Unlike vanilla diffusion, planned diffusion does not generate a predetermined number of tokens, so the number of denoising steps cannot be fixed. We define a parameter $r$ called the *steps ratio*. Given a generation length $|x|$, the number of denoising steps will be $s = r * |x|$. For a plan $z$ and multiple diffusion chunks lengths $l_1, ..., l_K$, the step ratio is defined as $s = r * \max_k l_k$. The number of denoising steps depends only on the length of the longest chunk, because tokens in each chunk are denoised in parallel to other chunks. Higher step ratio corresponds to higher quality and slower generation, while lower step ratio leads to faster generation but worse quality.

## 5 EXPERIMENTAL EVALUATION

We experimentally assess the performance of planned diffusion, focusing on its trade-off between generation quality and latency. Our results show that planned diffusion expands the latency-quality Pareto frontier for text generation when compared to autoregressive and other diffusion-based approaches. Furthermore, we demonstrate that our method scales better with additional compute; planned diffusion continues to improve with more training, whereas the performance of the autoregressive baseline plateaus.

**Training setup.** We fine-tune Dream-7B-Base (Ye et al., 2025; Qwen et al., 2024); the base model is first pre-trained autoregressively and then further pre-trained with a diffusion objective. We train with AdamW (Kingma & Ba, 2015; Loshchilov & Hutter, 2019), peak learning rate $5 \times 10^{-5}$ with linear decay, and bfloat16 precision. We use per-GPU batch size 1 and global batch size 4. Because autoregressive and diffusion language models have different optimal epoch counts (Prabhudesai et al., 2025), we sweep epochs over $\{2, 4, 8, 16\}$. We fine-tune on Gemini-annotated SlimOrca instruction-following data (see Section 4.2); we keep the control tags for planned diffusion and strip them for autoregressive and diffusion baselines. Training runs on $4 \times$H200 (141 GB) with PyTorch (Imambi et al., 2021) and HuggingFace (Wolf et al., 2020).

**Decoding strategies.** We compare seven decoding strategies. (1) *Autoregressive* samples tokens sequentially from the autoregressive model. (2) *Diffusion* samples masked tokens in parallel from the diffusion model; we configure the number of denoising steps to equal the number of new tokens as this produces the highest quality generation (Lou et al., 2024; Shi et al., 2024; Sahoo et al., 2024). (3) *Fast-dLLM* (Wu et al., 2025) samples from the same diffusion model but with an inference-time only optimization. We configure denoising steps to be half the number of new tokens and use a confidence threshold of 0.9, which Wu et al., 2025 use as the default value. (4) *Pasta-SFT (Jin et al., 2025)* utilizes control tags to implement semantic parallelism in a purely autoregressive setting. We compare to the SFT-trained version of Pasta, as opposed to RL-trained, for fair comparison with planned diffusion, which is trained only with SFT. (5) *Skeleton-of-Thought* (Ning et al., 2024) autoregressively samples a bullet-point outline, then applies regular-expression-based syntactic pattern matching for extracting points for parallel expansion. (6) *Planned diffusion* (ours) samples a denoising plan autoregressively from the planned diffusion model, then samples masked tokens within each chunk in parallel from the same model; for each chunk, we configure denoising steps to equal its predicted token count. (7) *Planned diffusion with sparse attention* (ours) is a variant of planned diffusion. PDSA treat concurrently generated chunks as fully independent conditioned on the plan, and as such, denoises the chunks using block-sparse attention (c.f., Figure 6). By using

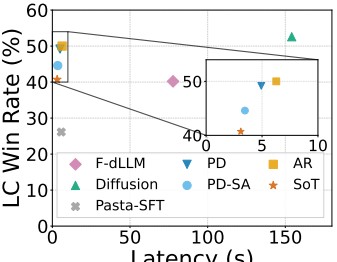 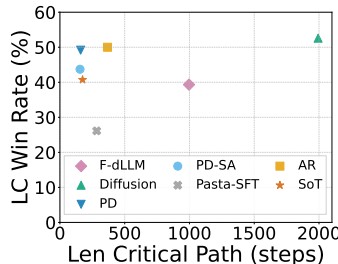 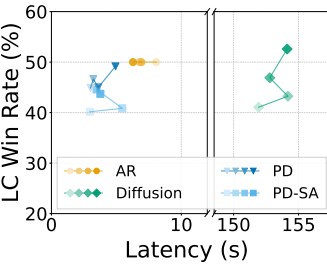

Figure 2: Evaluation of planned diffusion on the AlpacaEval benchmark. **Left:** A comparison of latency versus length-controlled win rate shows planned diffusion establishing a new Pareto frontier, offering a better trade-off between speed and quality. **Middle:** An analysis of the average critical path length reveals that planned diffusion requires substantially fewer sequential forward passes than autoregressive generation. **Right:** A scaling analysis shows that planned diffusion's win rate continues to improve with more training, while the autoregressive baseline's performance flatlines. Within each method, color brightness from lightest to darkest encodes 2, 4, 8, and 16 traing epochs. **AR**: Autoregressive; **F-dLLM**: Fast-dLLM; **SoT**: Skeleton-of-Thought; **PD**: Planned Diffusion; **PDSA**: Planned Diffusion Sparse Attention.

a sparse attention mask, PDSA improves compute efficiency with sparse attention implementations but reduces model expressivity and GPU utilization from vanilla planned diffusion.

**Inference setup.** We sample with temperature $0.2$ and top-$p$ $0.95$ following (Ye et al., 2025), and cap the total sequence length at $2048$ tokens. Inference runs on the same H200 hardware configuration.

**Benchmark and metrics.** We evaluate on AlpacaEval (805 instruction-following prompts) (Li et al., 2023; Dubois et al., 2024). For each method we report: (i) *average latency*—the mean wall-clock time per response; and (ii) *quality*—length-controlled win rate (LCWR) with an LLM-as-judge. We use the recommended default configuration from (Dubois et al., 2024) due to its high correlation with human preference. We fix the LCWR reference to the best autoregressive baseline. We identify this reference by evaluating the autoregressive variants (2, 4, 8, and 16 epochs) against the 2-epoch variant and choosing the model with the highest quality [1]. The 16-epoch model wins and serves as the fixed reference for all LCWR scores.

**Quality and Latency.** We plot the latency–quality trade-off in the left figure of Figure 2. Planned diffusion (PD) achieves a 22.4× speedup over Fast-dLLM (F-dLLM) and higher quality (44.6% vs. 40.2% length-controlled win rate). Relative to autoregressive decoding (AR), planned diffusion achieves a 1.27× speedup while achieving a 49.2% length-controlled win rate against the autoregressive reference (50.0%). Planned diffusion with sparse attention (PDSA) provides another point on the speed-quality Pareto frontier by further improving speedup to 1.81× relative to autoregressive decoding at 44.6% win rate. Diffusion attains the highest quality (52.6%) but requires substantially more inference time, requiring 25× the latency of autoregressive decoding. Compared to other semantic parallelism methods, planned diffusion is firmly on the Pareto frontier. Skeleton-of-thought offers marginally faster generations but incurs a significant decrease in accuracy. Pasta-SFT fails to properly learn the control tag semantics without additional RL training and produces worse quality and latency.

**Speedup Analysis.** We attribute much of planned diffusion's speedup over autoregressive decoding to its shorter *critical path* of generation. We define critical path length as the number of forward passes required to produce the final response. In a dLLM, the critical path will be given by the number of denoising steps $s$, so the total complexity is the cost of attention multiplied by the number iterations: $O(n^2 * s)$, where $n$ is the sequence length. Planned diffusion reduces the number of steps $s$ by a factor of $l_{max}/n$, where $l_{max}$ is the length of the longest chunk within a plan. The middle panel of Figure 2 shows that, on AlpacaEval, the average critical path of autoregressive decoding is 2.3× as long as that of planned diffusion (PD, 367.3 vs. 160.0 steps) and 2.8× as long as the sparse attention variant (PDSA, 367.3 vs. 155.2 steps). This is expected, as planned diffusion enables multiple chunks to denoise simultaneously. The realized speedup (1.85×) is smaller than the critical

---

[1]They tie on length controlled win rate, so we break ties using raw win rates.

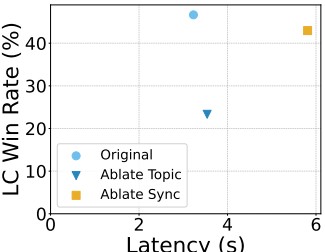 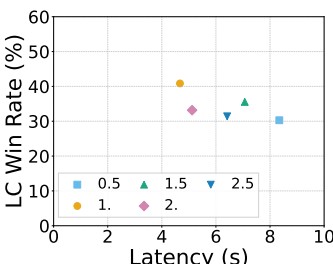 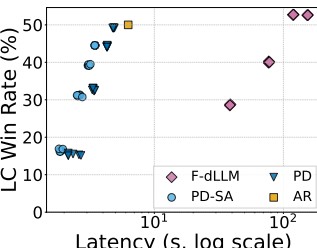

Figure 3: Additional analysis. **Left (Plan ablation):**Removing `topic` from autoregressive plan harms quality yet removing `<sync/>` tokens boosts speedup without significantly hurting quality. **Middle (chunk lengths):** LCWR peaks at a length-scaling factor of 1.0; length prediction in autoregressive plan does not exhibit systemic error. **Right (Quality versus Latency Sweep):** Varying the step ratio ($r = \{0.25, 0.5, 0.75, 1\}$) and the confidence threshold ($\tau = \{0.4, 0.5, 0.6, 0.7, 0.8, 0.9\}$) hyperparameters produces a smooth quality-latency trade-off.

path reduction (2.8× for PD, 2.3× for PD-DA) because each planned diffusion step does more work: KV-cache reuse is lower and per-step compute is heavier than an autoregressive token step.We also observe a difference in the total number of tokens generated (this includes control tokens and pad tokens) between planned diffusion and autoregressive decoding. Planned diffusion and planned diffusion with sparse attention produce approximately 9.1%[2] and 3.4% fewer tokens, respectively.

**Scaling.** The right panel of Figure 2 shows how the latency-quality tradeoff evolves with training epochs for the four training configurations we examine.

Autoregressive training shows no benefit from additional training compute: the length-controlled win rate remains flat at 50.0% across 2, 4, 8, and 16 epochs. Both variants of planned diffusion benefit moderately from additional training compute. Planned diffusion improves from 44.9% (2 epochs) to 49.2% (16 epochs), a gain of 4.3 percentage points, while planned diffusion with sparse attention improves from 40.2% (2 epochs) to 43.7% (16 epochs), a gain of 3.5 percentage points. Diffusion benefits significantly from additional training compute, rising from 41.1% (2 epochs) to 52.6% (16 epochs), a gain of 11.5 percentage points that ultimately surpasses the autoregressive baseline.

**Takeaway.** Planned diffusion sets a new latency–quality Pareto frontier and continuously improves with more training, while the autoregressive baseline plateaus.

## 6 ADDITIONAL ANALYSIS

We present additional analysis on several design decisions key to planned diffusion.

**Denoising Plan Ablation.** We remove two components of the autoregressive plan and evaluate their impact on inference quality and latency. We train all model variants for 4 epochs in this study. The original 4-epoch planned diffusion achieves an LC win rate of 46.65% at 3.23 s latency. Removing the `topic` attribute from the training data and re-training the planned diffusion model significantly harms inference quality. Specifically, it reduces the length controlled win rate from 46.65% to 23.33% and the latency from 3.23 seconds to 3.54 seconds. We conclude that `topic` attributes are critical for maintaining inference quality.

Recall that `<sync/>` marks a synchronization barrier: decoding beyond it begins only after all prior content is finalized. Deleting all `<sync/>` tokens from the training data and re-training the planned diffusion model modestly reduces quality. Specifically, the LC win rate falls from 46.65% to 42.96% while the latency increases from 3.23 s to 5.81 s. We conclude that omitting `<sync/>` slightly reduces quality but does not offer a latency benefit in this setting.

---

[2]The gap in output length is in part due to difference in optimal training epochs for planned diffusion (8ep) and autoregressive (16ep). The output length difference between 16ep planned diffusion and autoregressive reduces to 5.6%.

**Chunk Lengths.** We test whether the model predicts chunk lengths accurately. Accurate length prediction is key to achieving good generation quality for planned diffusion, as the diffusion denoising phase of generation cannot alter the chunk length. Systematic over-prediction wastes time by adding masks and denoising steps, while systematic under-prediction harms quality by forcing content truncation. To test for potential systematic deviation from the optimal generation length, we multiply the model's predicted chunk length by a length scaling factor to set the number of masks, sweeping the factor over $\{0.5, 1.0, 1.5, 2.0, 2.5\}$. We then measure length-controlled win rate (LCWR) and latency under identical inference settings. Latency rises with factors above $1.0$ as expected because larger chunks require more mask tokens and denoising steps. Quality peaks at the model's originally predicted length (i.e scaling factor of $1.0$). Deviating by $\pm 50\%$ of the predicted length reduces LCWR. Interestingly, we do not observe additional denoising steps leading to accuracy improvements. The model's length predictions are accurate; we do not observe systematic over/under-prediction.

**Quality versus Latency Sweep.** By default, to maximize generation quality, we set the number of denoising steps for a chunk equal to that chunk's length, and we unmask a position only when the model is highly confidence of its prediction. In this analysis, we measure the quality–latency trade-off by varying the number of denoising steps as well as confidence threshold for decoding.

Planned diffusion employs a hyperparameter called step ratio (see Section 4.2), which determines the number of denoising steps relative to the chunk length. The step ratio $r$ scales steps relative to chunk length. When $r$ equals 1 the number of steps equals the number of tokens; smaller $r$ uses fewer steps per token. The confidence threshold $\tau$ used by Wu et al. (2025) selects which positions to unmask at each denoising step. For each masked position, if the model's top-token probability at that position is at least $\tau$, we decode that token and unmask the position; otherwise we keep it masked for decoding at a later time step.

Sweeping $r$ over $\{0.25, 0.5, 0.75, 1\}$ and $\tau$ over $\{0.4, 0.5, 0.6, 0.7, 0.8, 0.9\}$ yields a smooth trade-off between generation quality and inference latency; Across the step ratio and the confidence threshold, planned diffusion and its sparse attention variant yield higher quality at equal or lower latency for most operating points.

**Takeaway.** Our analysis validates that our planning mechanism is minimal and reliable. At inference time, the step ratio and confidence threshold provide tunable control of the speed–quality trade-off.

# 7 CONCLUSION

In this work, we introduce planned diffusion, a novel hybrid architecture combining autoregressive planning with parallel diffusion-based execution to improve the trade-off between latency and quality in text generation. Our model exploits opportunities for parallelism within the semantic structure of text. Experimental evaluation shows that planned diffusion expands the latency–quality Pareto frontier, achieving a significant speedup over pure autoregressive and diffusion baselines with a minimal drop in quality. This efficiency gain is driven by a shorter critical path and our approach's scalability with respect to training. We also show planned diffusion offers fine-grained control over the speed-quality trade-off at inference time. Planned diffusion provides a promising framework for developing faster and more efficient large language models.

ACKNOWLEDGEMENT

The authors acknowledge the MIT Office of Research Computing and Data for providing high performance computing resources that have contributed to the research results reported within this paper. This work was supported in part by the MIT-Google Program for Computing Innovation and by SRC JUMP 2.0 (CoCoSys). This material is based upon work supported in part by the National Science Foundation Graduate Research Fellowship under Grant No. 2141064. Any opinions, findings, and conclusions or recommendations expressed in this material are those of the authors and do not necessarily reflect the views of the National Science Foundation. This work was funded in part by the DARPA ANSR, CODORD, and SAFRON programs under awards FA8750-23-2-0004, HR00112590089, and HR00112530141, NSF grant IIS1943641, and gifts from Adobe Research, Cisco Research, Qualcomm, and Amazon. This work was also supported by NSF CAREER Grant #2341040, Schmidt Sciences Early Career Fellowship, Okawa Foundation Research Award. Approved for public release; distribution is unlimited.

REPRODUCIBILITY STATEMENT

We specify the starting checkpoint, fine-tuning hyperparameters, inference settings, and hardware/software setup in Section 5. Section A includes a shortened version of the data-annotation prompt; we omit in-context examples for brevity.

LLM USE DISCLOSURE

We used LLMs to improve sentence-level writing and to search for related work.

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

## A  DATA ANNOTATION PROMPT

We present a shortened version of our data-annotation prompt below. We use it to instruct Gemini Flash 2.0 (temperature = 1.0, top-p = 0.95) to annotate our training data.

```
You will first identify whether the given chatbot response may be generated in
parallel. You are to then annotate the chatbot response using specific tags
that highlight segments suitable for parallel generation.

Use <async> tags to denote segments of text that may be generated asynchronously
in parallel with respect to the text that follows. Thus apply <async> tags only
to sentences that do not serve as necessary context for subsequent sentences.
Sentences that are crucial for understanding or generating following text are
not suitable for parallel asynchronous generation. For each <async> tag, include
a very concise topic description of the text surrounded within the <async> tags.
The topic description will be accessible to text generation after the closing
async tag to ensure continuity and coherence.

Use the singleton <sync/> tag for synchronization. All content generated before
<sync/>, including text marked by <async> is accessible to subsequent text
generation after the <sync/> tag, ensuring continuity and coherence.

Detailed Instructions:
- Tagging Rules:
 - Use <async> tag in pairs.
 - Ensure all text content is encapsulated within <async> tags.
 - Ensure that each <async> tag encompasses at least five words.
 - Refrain from altering the content of the response during annotation.
 - Use a maximum of 3 words in the topic description.
 - Use <sync/> sparingly as it introduces significant slowdown.
```

## B  TABLE SUMMARY OF EVALUATIONS

For completeness, we provide a table summary of the results in Figure 2.

Table 1: Comparison of methods based on latency and length-controlled win rate.

| Method | Latency (s) | LC Win Rate (%) |
|---|---|---|
| Diffusion | 154.124 | 52.59% |
| AR | 6.272 | 50.00% |
| Fast-dLLM Wu et al. (2025) | 77.723 | 40.20% |
| Pasta-SFT (Jin et al., 2025) | 5.740 | 26.13% |
| Skeleton-of-thought (SoT) (Ning et al., 2024) | 3.101 | 40.72% |
| Planned Diffusion | 4.934 | 49.17% |
| Planned Diffusion Sparse Attention (PDSA) | 3.471 | 44.59% |

## C  FULL SPEEDUP DISTRIBUTION

This section details the performance variance observed across the evaluation dataset. Figure 4 illustrates the distribution of speedup ratios obtained by comparing planned diffusion against the autoregressive baseline. The dataset consists of $N = 803$ test samples. The analysis yields a mean speedup ratio of $3.08$ and the distribution exhibits a standard deviation of $4.10$. This significant variance relative to the mean suggests a right-skewed distribution, indicating that while the average performance gain is substantial, there exists a subset of cases where the proposed method achieves exceptionally high speedups.

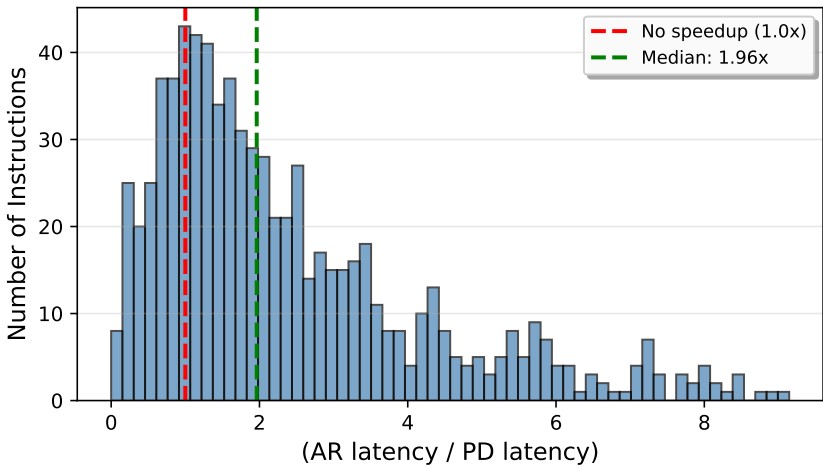

Figure 4: Distribution of Speedup Ratios.

## D  KV CACHING IN PLANNED DIFFUSION

KV caching plays a substantial role in the efficiency of planned diffusion. The KV cache of planned diffusion is derived from the model's hybrid attention mask. The general principle is that a token can be cached if its key and value embeddings are unaffected by future tokens. This is determined by whether a token attends to future positions in the sequence. The autoregressive planning stage uses causal attention and a conventional application of KV caching (Pope et al., 2023). In contrast, the diffusion stage employs a bidirectional mask, in which tokens inside an <async> chunk attend to each other. Because bidirectional attention does not support KV caching (Israel et al., 2025b), tokens inside an <async> chunk cannot be cached until their respective denoising process is complete. However, subsequent tokens, such as those in a new planning stage following a <sync/> tag, can efficiently attend to the KV cache of the preceding planning and diffusion stages. This caching mechanism is essential for combining the speed of diffusion with the computational savings of autoregressive KV caching.

## E  FURTHER DISCUSSION OF HYBRID MODELS

When referring to hybrid models between autoregression and discrete diffusion, we define a hybrid model as a single architecture, capable of pure left-to-right autoregression and pure discrete diffusion. This definition precludes existing works that interpolate between the autoregressive and discrete diffusion objective, such as Block Diffusion (Arriola et al., 2025), Eso-LM (Sahoo et al., 2025), or ARDMs Hoogeboom et al. (2022). Another necessary clarification is that discrete diffusion should not be confused with vanilla continuous diffusion. For continuous diffusion, multimodal hybrids over autoregrssion and diffusion exist, such as HybridVLA (Liu et al., 2025b) and Monoformer Zhao et al. (2024).

## F  DATASET PERFORMANCE BREAKDOWN

The AlpacaEval dataset Li et al. (2023) is composed of 5 datasets: self-instruct (Wang et al., 2023), open-assistant (Köpf et al., 2023), vicuna (Chiang et al., 2023), koala (Geng et al., 2023), and hh-rlhf (Bai et al., 2022). For planned diffusion, we report the length-controlled win rate for each and each standard error respectively. We observe that win rate is not especially sensitive to the dataset source of the instruction.

Table 2: Planned diffusion LC Win Rates across different datasets

| Dataset | Number of Samples | LC Win Rate (% $\pm$ SE) |
|---|---|---|
| self-instruct | 249 | 45.7 $\pm$ 0.65 |
| open-assistant | 187 | 44.2 $\pm$ 0.91 |
| vicuna | 80 | 44.2 $\pm$ 1.66 |
| koala | 156 | 48.0 $\pm$ 0.57 |
| hh-rlhf | 129 | 46.3 $\pm$ 1.04 |

## G  QUALITATIVE ANALYSIS

To better understand the behavior of Planned Diffusion, we analyze specific examples of generated outputs, focusing on the interaction between the autoregressive planning stage and the parallel diffusion stage. The examples below display the instruction and the raw output, which reveals the internal control tags generated by the planner.

In the raw output:

- `<promise>-<topic>... length</topic>` indicates the autoregressive plan for a specific span.

- `<async>...</async>` indicates the text generated in parallel via discrete diffusion.

- `[PAD]` tokens indicate the unused capacity of the allocated span length.

### G.1  SUCCESSFUL EXAMPLE

The following example demonstrates a successful decomposition of a list-based query. The instruction asks for popular careers in development studies. The autoregressive planner successfully identifies distinct semantic entities before generation begins.

The planner generates specific, discriminatory topics for each span: `UN orgs`, `WB orgs`, `NGOs`, `media orgs`, etc. Because the planning stage explicitly disambiguates the content of each span, the subsequent diffusion execution stage, which generates these spans in parallel with independent attention masks, does not suffer from mode collapse. Each diffusion process is conditioned on a unique semantic target, resulting in a comprehensive, non-repetitive list that correctly follows the instruction.

---

**Instruction**

Please answer the following question based on the information provided in the article.

Development studies is an interdisciplinary branch of social science. Development studies is offered as a specialized master's degree in a number of reputed universities around the world. It has grown in popularity as a subject of study since the early 1990s, and has been most widely taught and researched in developing countries and countries with a colonial history, such as the UK, where the discipline originated. Students of development studies often choose careers in international organisations such as the United Nations, World Bank, non-governmental organisations (NGOs), media and journalism houses, private sector development consultancy firms, corporate social responsibility (CSR) bodies and research centers.

What are some popular careers for students of development studies

---

**Raw Output**

```
<promise>-<topic>introduction 2</topic>
<async>Some popular careers for students of development studies include:[PAD][PAD]
[PAD][PAD][PAD][PAD][PAD][PAD][PAD][PAD]</async><sync/>
<promise>-<topic>UN orgs 4</topic>
<promise>-<topic>WB orgs 4</topic>
<promise>-<topic>NGOs 4</topic>
<promise>-<topic>media orgs 4</topic>
<promise>-<topic>development consultancy 4</topic>
<promise>-<topic>CSR bodies 4</topic>
<promise>-<topic>research centers 4</topic>
<async>1. International Organizations: Students often work for organizations such as
the United Nations (UN) to address global issues, including poverty, education,
healthcare, and education.[PAD][PAD][PAD][PAD][PAD][PAD][PAD]</async>
<async>2. World Bank (WB): Similar to the UN, the World Bank offers opportunities to
work on projects related to development, finance, and infrastructure in various
countries.[PAD][PAD][PAD][PAD][PAD][PAD][PAD]</async>
<async>3. Non-governmental Organizations (NGOs): Many students of development studies
pursue careers in NGOs, as these organizations are often involved in development
projects and initiatives.[PAD][PAD][PAD][PAD][PAD][PAD][PAD][PAD]</async>
<async>4. Media and Journalism Houses: Students may become reporters and journalists,
covering topics related to global issues, development, and the work of various
organizations.[PAD][PAD][PAD][PAD][PAD][PAD][PAD][PAD][PAD][PAD]</async>
<async>5. Private sector development consultancy firms: Companies that provide advice
and support to local businesses, helping them improve their operations, strategies,
and adapt to the market.[PAD][PAD][PAD][PAD][PAD][PAD][PAD][PAD]</async>
<async>6. Corporate Social Responsibility (CSR) bodies: Students can work in these
bodies to help businesses and organizations understand and meet their social
responsibilities and promote sustainable practices.[PAD][PAD][PAD][PAD][PAD][PAD][PAD]
[PAD]</async>
<async>7. Research Centers: Students may also work in research centers, where they can
 contribute to the advancement of development studies research, analysis, and
presentation.[PAD][PAD][PAD][PAD][PAD][PAD][PAD][PAD][PAD][PAD]</async><sync/>
<promise>-<topic>summary 4</topic>
<async>These are just a few examples of the many careers available to students of
development studies. The field is diverse and offers opportunities to contribute to
the global advancement of development and social change.[PAD][PAD][PAD][PAD]</async>
```

---

**Clean Output**

Some popular careers for students of development studies include:

1. International Organizations: Students often work for organizations such as the United Nations (UN) to address global issues, including poverty, education, healthcare, and education.

2. World Bank (WB): Similar to the UN, the World Bank offers opportunities to work on projects related to development, finance, and infrastructure in various countries.

3. Non-governmental Organizations (NGOs): Many students of development studies pursue careers in NGOs, as these organizations are often involved in development projects and initiatives.

4. Media and Journalism Houses: Students may become reporters and journalists, covering topics related to global issues, development, and the work of various organizations.

5. Private sector development consultancy firms: Companies that provide advice and support to local businesses, helping them improve their operations, strategies, and adapt to the market.

6. Corporate Social Responsibility (CSR) bodies: Students can work in these bodies to help businesses and organizations understand and meet their social responsibilities and promote sustainable practices.

7. Research Centers: Students may also work in research centers, where they can contribute to the advancement of development studies research, analysis, and presentation. These are just a few examples of the many careers available to students of development studies. The field is diverse and offers opportunities to contribute to the global advancement of development and social change.

---

## G.2 FAILED EXAMPLE

The following example illustrates a failure mode that can result from poor planning. In this case, the autoregressive planner fails to decompose the semantic content effectively. Instead of generating specific locations (e.g., "NYC", "Denver", "LA"), the planner generates generic structural tags: `list part 1`, `list part 2`, `list part 3`.

---

**Instruction**

Make a list of the top places in the U.S. to visit at the given time.

November

---

**Raw Output**

```
<promise>-<topic>introduction 2</topic>
<promise>-<topic>list prompt 1</topic>
<async>Sure, here's a list of the top places in the U.S. to visit in November:</async>
<async>Here are the top places to visit:[PAD][PAD]</async><sync/>
<promise>-<topic>list part 1 4</topic>
<promise>-<topic>list part 2 3</topic>
<promise>-<topic>list part 3 3</topic>
<promise>-<topic>list part 4 3</topic>
<promise>-<topic>list part 5 3</topic>
<promise>-<topic>list part 6 3</topic>
<promise>-<topic>list part 7 3</topic>
<promise>-<topic>list part 8 3</topic>
<promise>-<topic>list part 9 3</topic>
<promise>-<topic>list part 10 3</topic>
<async>1. New York City[PAD][PAD][PAD][PAD][PAD]</async>
<async>2. New York City - Lots of shopping, museums, and entertainment.[PAD][PAD][PAD]
[PAD][PAD]</async>
<async>2. New York City - It's a big city with lots of fun things, shops, and places
to eat.[PAD][PAD][PAD][PAD][PAD][PAD]</async><async>3. New York City - New York City
is a big city with lots of things to see, like the Statue of Liberty, the park, and
the buildings.[PAD][PAD][PAD][PAD][PAD][PAD][PAD]</async>
<async>4. Denver, Colorado - A great place for football, hiking, and skiing. It has
the big city of Denver, the state of Colorado, and the beautiful mountains around it.
It's also famous for the Denver Zoo.[PAD][PAD][PAD][PAD]</async>
<async>5. Los Angeles, California - Known for its beautiful coastline, Hollywood, and
shopping, Los Angeles is a top place to visit in November. Visit places like Santa
Monica, Santa Barbara, and Santa Cruz, and enjoy the local food, beaches, and culture.
[PAD][PAD][PAD][PAD][PAD][PAD][PAD]</async>
<async>8. Sacramento, California - Known for its rich history and beautiful coastline,
 Sacramento is a major city in the state of California. It is the capital city of the
United States and is home to many major events and festivals, such as the Sacramento
Street Festival, the Sacramento Zoo, and the Sacramento Riverwalk.[PAD][PAD][PAD][PAD]
[PAD][PAD][PAD][PAD]</async>
<async>10. Denver, Colorado
- Denver is known for its beautiful landscapes, history, and art scene. It's one of
the most popular cities in the U.S. and has a lot to do and see. Visit the U.S. Space
Center, the Denver Zoo, and the University of Colorado Boulder to get a taste of the
city.[PAD][PAD][PAD][PAD][PAD][PAD][PAD][PAD][PAD][PAD]</async>
<async>10. Denver, Colorado
- Denver is a major city in the U.S. with diverse history, culture, and landscapes. It
 is home to numerous attractions, such as the Denver Zoo, Science Center, andUSS, as
well as the University of Colorado Boulder and the Denver Zoo. The city is also known
for its mountains, forests, and parks, making it a great place for outdoor activities.
[PAD][PAD][PAD][PAD][PAD][PAD][PAD]</async>
<async>10. Denver, Colorado
- Denver is a major city in the U.S. and offers a mix of history, culture, and beauty.
 Some of the top places to visit in Denver include the U.S. Open Center, Central Park,
 the Denver Zoo, and the Denver Nature Center. The University of Colorado, also
located in Denver, is a great place to explore, and the Denver Art Museum offers a
unique glimpse into the city.[PAD][PAD][PAD][PAD][PAD][PAD][PAD][PAD][PAD][PAD]</async
>
```

---

**Clean Output**

Sure, here's a list of the top places in the U.S. to visit in November: Here are the top places to visit:
1. New York City
2. New York City - Lots of shopping, museums, and entertainment.
2. New York City - It's a big city with lots of fun things, shops, and places to eat.
3. New York City - New York City is a big city with lots of things to see, like the Statue of Liberty, the park, and the buildings.
4. Denver, Colorado - A great place for football, hiking, and skiing. It has the big city of Denver, the state of Colorado, and the beautiful mountains around it. It's also famous for the Denver Zoo.
5. Los Angeles, California - Known for its beautiful coastline, Hollywood, and shopping, Los Angeles is a top place to visit in November. Visit places like Santa Monica, Santa Barbara, and Santa Cruz, and enjoy the local food, beaches, and culture.
8. Sacramento, California - Known for its rich history and beautiful coastline, Sacramento is a major city in the state of California. It is the capital city of the United States and is home to many major events and festivals, such as the Sacramento Street Festival, the Sacramento Zoo, and the Sacramento Riverwalk.
10. Denver, Colorado
- Denver is known for its beautiful landscapes, history, and art scene. It's one of the most popular cities in the U.S. and has a lot to do and see. Visit the U.S. Space Center, the Denver Zoo, and the University of Colorado Boulder to get a taste of the city.
10. Denver, Colorado
- Denver is a major city in the U.S. with diverse history, culture, and landscapes. It is home to numerous attractions, such as the Denver Zoo, Science Center, andUSS, as well as the University of Colorado Boulder and the Denver Zoo. The city is also known for its mountains, forests, and parks, making it a great place for outdoor activities.
10. Denver, Colorado
- Denver is a major city in the U.S. and offers a mix of history, culture, and beauty. Some of the top places to visit in Denver include the U.S. Open Center, Central Park, the Denver Zoo, and the Denver Nature Center. The University of Colorado, also located in Denver, is a great place to explore, and the Denver Art Museum offers a unique glimpse into the city.

---

## H   PLANNED DIFFUSION ATTENTION MASKS

The attention mask for planned diffusion combines causal and bidirectional attention. Causal attention is used for sequential planning stages. Bidirectional attention is used within `<async>` chunks for parallel denoising. We enable concurrent chunks to cross-attend, unlike the sparse attention variant. After a `<sync/>` token, subsequent tokens can to attend to all prior tokens. For illustrative purposes, this example shortens the diffusion chunks.

In planned diffusion sparse attention (SPDA), the asynchronous blocks cannot attend to on another. Sparsity enables more compute efficient training and inference.

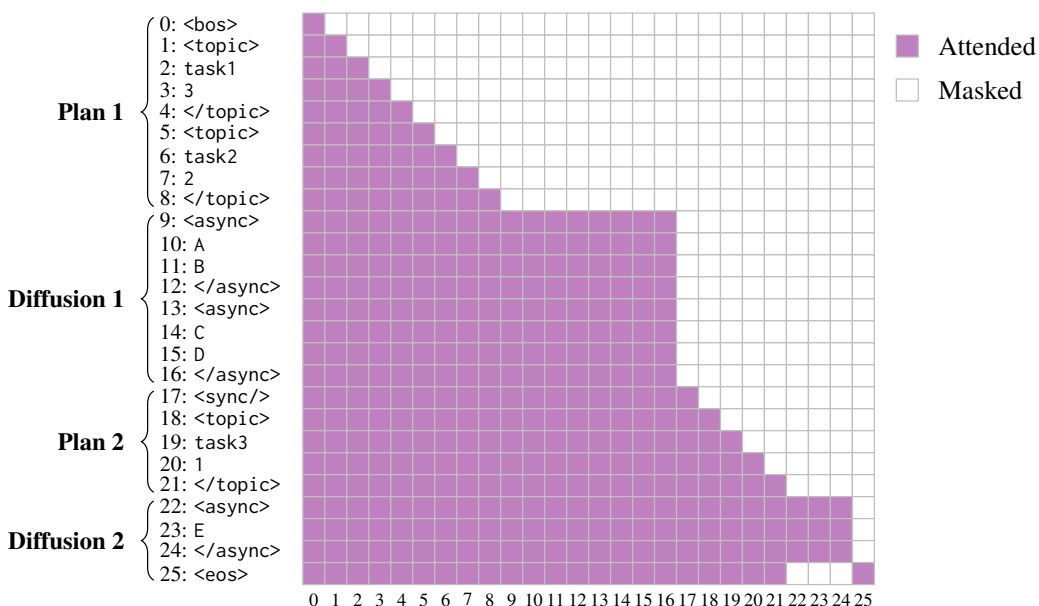

Figure 5: Attention mask for planned diffusion.

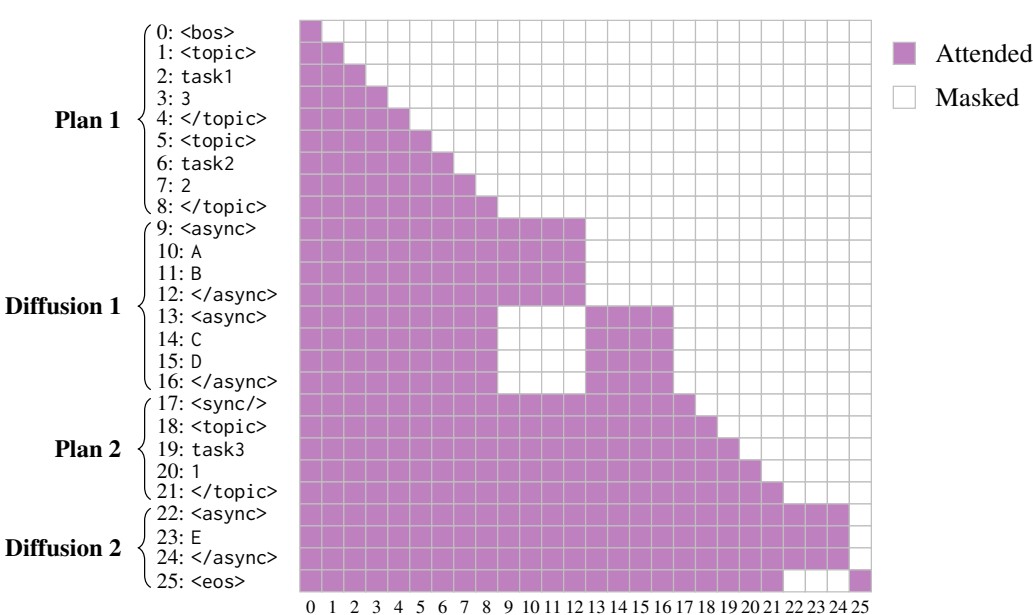

Figure 6: Attention mask for planned diffusion with sparse attention (PDSA).

