# OpenReview forum: "Planned Diffusion"
_ICLR.cc/2026/Conference — ICLR 2026 Poster_

### Official Review · Reviewer_Yrr4 · 2025-10-23

**Soundness:** 2
**Presentation:** 3
**Contribution:** 3
**Rating:** 6
**Confidence:** 3

**Summary:**

The paper proposes Planned Diffusion, a hybrid decoding framework that first generates a short autoregressive plan (with control tags like <async>…</async>, <sync/>, topic and predicted span length), then diffuses multiple spans in parallel under a bidirectional mask. It integrates KV-cache reuse across stages and uses an energy-ordered unmasking rule for diffusion. On AlpacaEval (805 prompts), it reports 1.27×–1.81× speedup over AR with 0.87%–5.4% LC win-rate drop, and shows ablations on plan components, span-length prediction, and quality–latency knobs (step-ratio r, confidence threshold τ).

**Strengths:**

# Strengths

1) Clear plan-then-parallel idea
- First produce a short AR plan, then decode multiple spans in parallel with a bidirectional mask.
- Turns semantic parallelism into practical decoding parallelism.

2) Simple, single-model pipeline
- Causal attention for planning; bidirectional attention within spans during diffusion.
- KV-cache reuse and stage transitions are straightforward to add to existing inference stacks.

3) Transparent quality–latency control
- Tunable knobs (e.g., diffusion-step ratio *r*, confidence threshold *τ*) expose a smooth speed–quality trade-off.
- Pareto curves make the effect of these knobs easy to interpret.

4) Useful diagnostics
- Component-wise ablations (control tags, span-length prediction, unmasking rule) clarify each module’s role.
- The explicit plan → execute interface helps surface failure cases.

5) Reasonable reproducibility details
- Checkpoints, fine-tuning settings, and prompt snippets are documented sufficiently for re-implementation.

**Weaknesses:**

# Weaknesses

1) Modest speedup and weaker quality
- End-to-end acceleration is limited (≈ **1.27×**).
- Reported quality can trail standard AR decoding (**49.2 vs. 50**).

2) Reliance on prompt-model data construction
- Plan annotations depend on a separate prompt model, introducing data-generation overhead and potential bias.
- The dependence raises questions about scalability of the approach as model/data sizes grow.

3) Narrow experimental scope
- Evaluation is conducted on a **single benchmark**.
- Lacks comparisons against a broader set of **model baselines** and decoding accelerators.

4) Planning limitations on complex tasks
- For tasks with strong cross-span dependencies, plan segmentation and span sizing are uncertain.
- Observed speedup may be confounded by **implicit output-length constraints** rather than true parallelism.

**Questions:**

ref weakness

---

> ### Author Response · Authors · 2025-11-22
> **Response to Reviewer Yrr4 (1)**
>
> Dear Reviewer Yrr4,
>
> We thank the reviewer for the constructive feedback and for recognizing the clarity of our plan-then-parallel idea, the simplicity of our single-model pipeline, and the transparency of our quality-latency controls. We appreciate your assessment that the work stands above the acceptance threshold and we address your specific concerns below.
>
> (1) Modest speedup and weaker quality
>
> We wish to clarify that, as stated in our abstract, we report a speedup of 1.84x with a corresponding 6.8% drop in win rate. It is important to contextualize these results within the current landscape of non-autoregressive generation; diffusion language models have historically struggled to obtain any speedup over autoregressive models while maintaining comparable quality [8]. Our approach represents a vast improvement over vanilla dLLMs and effectively challenges autoregressive LLMs, which are widely regarded as the current state-of-the-art paradigm. Our objective is to push the Pareto frontier of this trade-off, offering a viable alternative where previous attempts have failed to balance latency and performance.
>
>
> (2) Reliance on prompt-model data construction.
>
> With respect to the concern regarding data generation overhead, we note that modern LLMs are frequently trained with Supervised Fine-Tuning (SFT) data that has been curated or generated by other LLMs [2]. We consider our work to be fully consistent with such established methodologies. In our case, we utilize synthetic data generation specifically to enhance parallelism and inference efficiency, rather than solely for the purpose of answer quality.
>
> (3) Narrow experimental scope
>
> We respectfully disagree with the characterization of our experimental scope as narrow. We note that Length-Controlled AlpacaEval is a robust metric known to possess a high correlation with human preferences derived from Chatbot Arena [1]. Furthermore, AlpacaEval [2] is not a monolithic task but a composite of five distinct datasets: Self-Instruct [3], Open-Assistant [4], Vicuna [5], Koala [6], and HH-RLHF [7]. This composition affords a holistic view of instruction-following capabilities across diverse settings. For instance, the Self-Instruct subset alone spans domains including email writing, social media, productivity tools, and programming (see Figure 3 of Wang et al. [3] for a complete taxonomy).
>
> In addition to autoregressive (AR), Diffusion, and Fast-dLLM baselines, we evaluate two prior semantic-parallel decoding methods: Pasta-SFT [9] and Skeleton-of-Thought (SoT, [10]). Pasta-SFT implements semantic parallelism via control tags in a purely autoregressive model and, in our setting, we use its SFT-only variant (without RL) for a fair comparison to planned diffusion. SoT, by contrast, first samples a bullet-point skeleton autoregressively and then applies regex-based parsing to extract points for parallel expansion. Empirically, both lag behind planned diffusion: SoT is slightly faster than PD (3.10 s vs. 3.47 s) but suffers a notable drop in length-controlled win rate (40.7 % vs. 44.6 %), while in the absence of further RL training, Pasta-SFT fails to learn the control-tag semantics and yields substantially worse quality (26.1 %) at higher latency (5.74 s).
>
> (4) Planning limitations on complex tasks
>
> Finally, concerning the uncertainty of plan segmentation on complex tasks, we emphasize that all evaluations were set to a 2K context length to ensure fairness across comparisons. We posit that the observed speedup is not confounded by implicit output-length constraints; rather, further speedup can be enabled without this context length limit. We anticipate that semantic parallelism likely increases with sequence length, allowing our method to capitalize on longer dependencies effectively.

---

> > ### Author Response · Authors · 2025-11-22
> > **Response to Reviewer Yrr4 (2)**
> >
> > [1] Dubois, Yann, et al. "Length-controlled alpacaeval: A simple way to debias automatic evaluators." arXiv preprint arXiv:2404.04475 (2024).
> >
> > [2] Li, Xuechen, et al. ‘AlpacaEval: An Automatic Evaluator of Instruction-Following Models’. GitHub Repository, GitHub, 5 2023, github.com/tatsu-lab/alpaca_eval.
> >
> > [3] Wang, Yizhong, et al. "Self-instruct: Aligning language models with self-generated instructions." Proceedings of the 61st annual meeting of the association for computational linguistics (volume 1: long papers). 2023.
> >
> > [4] Kopf, Andreas et al. “OpenAssistant Conversations - Democratizing Large Language Model Alignment.” ArXiv abs/2304.07327 (2023): n. Pag.
> >
> > [5] Chiang, Wei-Lin, et al. Vicuna: An Open-Source Chatbot Impressing GPT-4 with 90%* ChatGPT Quality. Mar. 2023, lmsys.org/blog/2023-03-30-vicuna/.
> >
> > [6] Geng, Xinyang, et al. Koala: A Dialogue Model for Academic Research. Apr. 2023, bair.berkeley.edu/blog/2023/04/03/koala/.
> >
> > [7] Bai, Yuntao, et al. "Training a helpful and harmless assistant with reinforcement learning from human feedback." arXiv preprint arXiv:2204.05862 (2022).
> >
> > [8] Peng, Baolin, et al. "Instruction tuning with gpt-4." arXiv preprint arXiv:2304.03277 (2023).
> >
> > [9] Jin, Tian, et al. "Learning to keep a promise: Scaling language model decoding parallelism with learned asynchronous decoding." Proceedings of the 42nd International Conference on Machine Learning (ICML). 2025.
> >
> > [10] Ning, Xuefei, et al. "Skeleton-of-Thought: Prompting LLMs for efficient parallel generation." Proceedings of the Twelfth International Conference on Learning Representations (ICLR). 2024.

---

> > > ### Comment · Reviewer_Yrr4 · 2025-11-27
> > >
> > > Thank you for your response — it has addressed most of my concerns. At this point, I’m inclined to keep my current score.

---

### Official Review · Reviewer_Zq7e · 2025-10-30

**Soundness:** 3
**Presentation:** 3
**Contribution:** 3
**Rating:** 6
**Confidence:** 3

**Summary:**

This paper introduces planned diffusion, a hybrid text generation method combining autoregressive planning with diffusion-based parallel execution. The model first generates a structured plan with control tags defining independent text spans, then generates these spans simultaneously via discrete diffusion. Evaluated on AlpacaEval, the method achieves 1.84× speedup over autoregressive generation with a 6.8% drop in win rate, establishing a new point on the latency-quality Pareto frontier.

**Strengths:**

1. First text-only model combining discrete diffusion with autoregression in a unified architecture, addressing the speed-quality tradeoff from a novel angle
2. Hybrid attention masking elegantly enables both causal and bidirectional attention; KV caching strategy is well-designed for this architecture
3. Establishes new Pareto frontier point; sensitivity analysis confirms model learns accurate length prediction without systematic bias
4. Method is orthogonal to other acceleration techniques and continues improving with more training data

**Weaknesses:**

1. Only AlpacaEval benchmark; no evaluation on diverse tasks (summarization, QA, code generation, creative writing). How does performance vary across task types?
2. No direct comparison to other semantic parallelism methods (e.g., Skeleton-of-Thought, APAR, ParaThinker) despite extensive related work discussion. This is critical for establishing true contribution.
3. Relies on Gemini for training data annotation. What is annotation quality? How many examples were rejected? Could this be learned end-to-end without synthetic supervision?

**Questions:**

1. How does planned diffusion compare quantitatively to other semantic parallelism methods?

2. How does performance vary across task types beyond instruction-following (e.g., summarization, code generation, creative writing)?

3. What is the speedup variance across examples? Are there cases where planning overhead makes it slower than baseline?

4. What percentage of generations have poor plans? Can you provide failure case examples and error analysis?

5. What content types decompose well vs. poorly? Does the method struggle with sequential reasoning or narratives?

---

> ### Author Response · Authors · 2025-11-22
> **Response to Reviewer Zq7e (1)**
>
> Dear Reviewer Zq7e,
>
> We sincerely thank you for your thoughtful review and for highlighting that our method establishes a new point on the latency-quality Pareto frontier. We appreciate your recognition of our hybrid attention masking as an elegant solution for enabling both causal and bidirectional attention, and your assessment that Planned Diffusion addresses the speed-quality tradeoff from a novel angle. We value your constructive feedback and address your specific questions below.
>
> (1) Only AlpacaEval benchmark
>
> Regarding the evaluation scope, we respectfully highlight that Length-Controlled AlpacaEval is a robust metric known to possess a high correlation with human preferences derived from Chatbot Arena [1]. Furthermore, AlpacaEval [2] is not a single task but a composite of five distinct datasets: Self-Instruct [3], Open-Assistant [4], Vicuna [5], Koala [6], and HH-RLHF [7]. We note that the Self-Instruct dataset specifically encompasses the diverse tasks you listed, including summarization, QA, code generation, and creative writing.
>
> (2) No direct comparison to other semantic parallelism methods
> In addition to autoregressive (AR), Diffusion, and Fast-dLLM baselines, we evaluate two prior semantic-parallel decoding methods: Pasta-SFT [9] and Skeleton-of-Thought (SoT, [10]). Pasta-SFT implements semantic parallelism via control tags in a purely autoregressive model and, in our setting, we use its SFT-only variant (without RL) for a fair comparison to planned diffusion. SoT, by contrast, first samples a bullet-point skeleton autoregressively and then applies regex-based parsing to extract points for parallel expansion. Empirically, both lag behind planned diffusion: SoT is slightly faster than PD (3.10 s vs. 3.47 s) but suffers a notable drop in length-controlled win rate (40.7 % vs. 44.6 %), while in the absence of further RL training, Pasta-SFT fails to learn the control-tag semantics and yields substantially worse quality (26.1 %) at higher latency (5.74 s).
>
> (3) Relies on Gemini for training data annotation.
>
> We analyzed our training data and found that out of 96,407 total samples, 95,422 were valid, resulting in a validity fraction of 0.990. This indicates that only 1% of examples were rejected, demonstrating high annotation quality. We consider it unlikely that semantic parallelism can be effectively learned without such synthetic supervision, as identifying semantic independence requires the model to understand the text at a high level. We note that instruction-following LLMs are routinely trained with vast amounts of synthetic data [8], suggesting that training on synthetic data is within standard practice and not a significant drawback.

---

> ### Author Response · Authors · 2025-11-22
> **Response to Reviewer Zq7e (2)**
>
> **Questions**
>
> > (1) How does planned diffusion compare quantitatively to other semantic parallelism methods?
>
> Please refer to our answer above for additional comparisons to prior semantic parallelism methods.
>
> > (2) How does performance vary across task types beyond instruction-following (e.g., summarization, code generation, creative writing)?
>
> As previously explained, AlpacaEval actually encompasses all such tasks (summarization, code generation and creative writing) and considers them all to be instruction-following tasks. In Appendix F, we present a breakdown of win rate across five different subsets of AlpacaEval and observe that win rate is reasonably consistent across five dataset subsets.
>
> > (3) What is the speedup variance across examples? Are there cases where planning overhead makes it slower than baseline?
>
> We present a visual characterization of speedup distributions across hundreds of responses we evaluate in newly updated Appendix C. And yes, while there are rare cases where planned diffusion is slower than baseline; however, overall we observe compelling speedups with respect to autoregressive baseline.
>
> > (4) What percentage of generations have poor plans? Can you provide failure case examples and error analysis?
>
> While we believe that planning errors are implicitly captured within the overall evaluation scores, we agree that qualitative analysis is valuable for understanding the model's behavior. We have therefore included specific examples of failure cases in the "Qualitative Analysis" section of the Appendix in our revised manuscript to provide the requested error analysis.
>
> > (5) What content types decompose well vs. poorly? Does the method struggle with sequential reasoning or narratives?
>
> We have not established a precise taxonomy of task performance across all possible content types. However, given that our current setup necessitates semantic independence between text spans to maintain performance, we anticipate that the method may face challenges on tasks that feature inherent sequential dependence, such as strict narrative generation or multi-step reasoning. We do not view this difficulty as intractable, but rather we identify the handling of strict sequential dependencies as a key direction for future research in the field of semantic parallelism.
>
>
> [1] Dubois, Yann, et al. "Length-controlled alpacaeval: A simple way to debias automatic evaluators." arXiv preprint arXiv:2404.04475 (2024).
>
> [2] Li, Xuechen, et al. ‘AlpacaEval: An Automatic Evaluator of Instruction-Following Models’. GitHub Repository, GitHub, 5 2023, github.com/tatsu-lab/alpaca_eval.
>
> [3] Wang, Yizhong, et al. "Self-instruct: Aligning language models with self-generated instructions." Proceedings of the 61st annual meeting of the association for computational linguistics (volume 1: long papers). 2023.
>
> [4] Kopf, Andreas et al. “OpenAssistant Conversations - Democratizing Large Language Model Alignment.” ArXiv abs/2304.07327 (2023): n. Pag.
>
> [5] Chiang, Wei-Lin, et al. Vicuna: An Open-Source Chatbot Impressing GPT-4 with 90%* ChatGPT Quality. Mar. 2023, lmsys.org/blog/2023-03-30-vicuna/.
>
> [6] Geng, Xinyang, et al. Koala: A Dialogue Model for Academic Research. Apr. 2023, bair.berkeley.edu/blog/2023/04/03/koala/.
>
> [7] Bai, Yuntao, et al. "Training a helpful and harmless assistant with reinforcement learning from human feedback." arXiv preprint arXiv:2204.05862 (2022).
>
> [8] Peng, Baolin, et al. "Instruction tuning with gpt-4." arXiv preprint arXiv:2304.03277 (2023).
>
> [9] Jin, Tian, et al. "Learning to keep a promise: Scaling language model decoding parallelism with learned asynchronous decoding." Proceedings of the 42nd International Conference on Machine Learning (ICML). 2025.
>
> [10] Ning, Xuefei, et al. "Skeleton-of-Thought: Prompting LLMs for efficient parallel generation." Proceedings of the Twelfth International Conference on Learning Representations (ICLR). 2024.

---

### Official Review · Reviewer_UEZx · 2025-11-01

**Soundness:** 3
**Presentation:** 3
**Contribution:** 4
**Rating:** 6
**Confidence:** 3

**Summary:**

The paper proposes Planned Diffusion, a hybrid text generation approach that first plans a response autoregressively (producing structure/length tags) and then fills multiple spans in parallel via discrete diffusion, aiming to shift the latency–quality Pareto frontier. On AlpacaEval, the method reportedly delivers ~1.8× speedup with a modest quality drop versus autoregressive decoding, and includes sensitivity analyses on span-length scaling and denoising step ratio.

**Strengths:**

- Clear, appealing idea: formal two-stage factorization (planning then parallel diffusion), with an explicit algorithm and attention-masking design.
- Well-specified control language (<topic>, <async>, <sync/>) that makes semantic parallelism concrete and implementable.
- Empirical evidence of a new speed/quality trade-off vs. AR and diffusion baselines (latency–quality plots, critical-path analysis, scaling behavior).
- Sensitivity analyses help demystify behavior: best performance when using the model’s predicted span lengths (scale=1.0) and a tunable quality–latency knob via step ratio.

**Weaknesses:**

- Benchmark scope: Results focus on AlpacaEval with an LLM-as-judge (LCWR). This is a useful proxy but not a robust test of coherence/faithfulness across diverse tasks (e.g., reasoning, long-form, safety). Lack of human evals or broader benchmarks (e.g., MT-Bench, GSM-8K reasoning slices, instruction-following suites) weakens generality.
- Baselines & fairness details: Diffusion is configured with steps equal to token count, and fast-dLLM with specific hyperparameters; however, broader ablations (other drafting/verification, semi-AR methods, SoTA speculative decoding stacks) are limited. This makes it harder to judge the absolute Pareto gains.
- Robustness & failure modes: The method assumes reliable plan quality (topic labels, span counts). What happens when planning is wrong (e.g., underestimates length; cross-span dependencies appear late)? The sensitivity section is a good start, but a qualitative error analysis is missing.
- Training data annotation relies on a proprietary LLM (Gemini) to insert tags under constraints. Potential concerns: annotation consistency, domain shift to noisier instructions, and whether the model overfits the tag scheme rather than learning general “semantic parallelism.” More diagnostics would help.

**Questions:**

- Plan robustness: How often does the planner significantly under/over-estimate span length in the wild? Could a lightweight “repair” step (e.g., local AR patching) recover quality when spans are misplanned?

- Generalization: Have you tried the approach on models with different pretraining (e.g., pure AR LLMs plus diffusion fine-tune) or at different scales?

- Error modes: Is there any qualitative examples where planned diffusion fails (e.g., subtle cross-span dependencies) and discuss mitigation.

---

> ### Author Response · Authors · 2025-11-22
> **Response to Reviewer UEZx (1)**
>
> Dear Reviewer UEZx,
>
> We thank Reviewer UEZx for their constructive feedback and for recognizing our method as a "clear, appealing idea" with a "well-specified control language." We are encouraged by the reviewer’s appreciation of our empirical evidence regarding the speed/quality trade-off and the utility of our sensitivity analyses. We address the specific concerns and questions below.
>
> Benchmark scope
>
> Regarding the evaluation scope, we respectfully note that Length-Controlled AlpacaEval is a robust metric known to possess a high correlation with human preferences derived from Chatbot Arena [1]. Furthermore, AlpacaEval [2] is not a single task but a composite of five distinct datasets: self-instruct [3], open-assistant [4], vicuna [5], koala [6], and hh-rlhf [7]. This composition provides a holistic view of instruction-following capabilities across diverse settings; for instance, the self-instruct subset alone spans domains including email writing, social media, productivity tools, and programming (see Figure 3 of Wang et al. [3] for a complete taxonomy). Of the tasks the reviewer mentioned, “reasoning” is included in self-instruct [3], “long-form” is included in vicuna [5], and “safety” is included in hh-rlhf [7]. We have included in the Appendix of the revised manuscript a breakdown of win rate over each dataset.
>
> Baselines & fairness details
>
> In addition to autoregressive (AR), Diffusion, and Fast-dLLM baselines, we evaluate two prior semantic-parallel decoding methods: Pasta-SFT [8] and Skeleton-of-Thought (SoT, [9]). Pasta-SFT implements semantic parallelism via control tags in a purely autoregressive model and, in our setting, we use its SFT-only variant (without RL) for a fair comparison to planned diffusion. SoT, by contrast, first samples a bullet-point skeleton autoregressively and then applies regex-based parsing to extract points for parallel expansion. Empirically, both lag behind planned diffusion: SoT is slightly faster than PD (3.10 s vs. 3.47 s) but suffers a notable drop in length-controlled win rate (40.7 % vs. 44.6 %), while in the absence of further RL training, Pasta-SFT fails to learn the control-tag semantics and yields substantially worse quality (26.1 %) at higher latency (5.74 s).
>
> Robustness & failure modes
>
> We appreciate the suggestion regarding robustness and failure modes and have added a section titled "Qualitative Analysis" to the Appendix of the revised manuscript. As noted in the text, a potential failure mode occurs when a plan lacks sufficient detail, causing independent spans to generate redundant content. However, quantifying plan-specific issues on a case-by-case basis remains challenging. We therefore maintain that the overall error analysis and the robustness of the method are best captured effectively by the aggregate performance metrics shown in Figure 3.
>
> Training data
>
> Regarding the concerns about training data annotation and potential overfitting to the tag scheme, we argue that the model’s strong downstream performance serves as evidence for the quality of our annotation scheme and dataset. If the model were merely overfitting to the syntax of the tags rather than learning the underlying semantic parallelism, it would fail to generate the coherent, high-quality answers required to achieve the high win rates observed in Figure 3.

---

> ### Author Response · Authors · 2025-11-22
> **Response to Reviewer UEZx (2)**
>
> **Questions**
>
> > Plan robustness: How often does the planner significantly under/over-estimate span length in the wild? Could a lightweight “repair” step (e.g., local AR patching) recover quality when spans are misplanned?
>
> Regarding plan robustness and the suggestion of a lightweight repair step, we emphasize that evaluating the quality of a plan generally requires observing the subsequent generation. Consequently, introducing an intermediate validation or repair step would inherently conflict with our primary objective of reducing latency. We argue that our competitive quality results serve as empirical evidence that the planner is robust and effectively captures semantic parallelism without requiring additional correction mechanisms.
>
> > Generalization: Have you tried the approach on models with different pretraining (e.g., pure AR LLMs plus diffusion fine-tune) or at different scales?
>
> Our current study restricts experiments to the 7B parameter scale to maintain a controlled environment for fair comparison. While we have not extended this to other model scales in this work, we have expanded our method-specific ablations. In the revised manuscript, we include an analysis of the attention mask design that allows for dependencies between spans, which yields higher quality responses (please see Figure 3 in the updated paper).
>
> > Error modes: Is there any qualitative examples where planned diffusion fails (e.g., subtle cross-span dependencies) and discuss mitigation.
>
> We refer the reviewer to the "Qualitative Analysis" section added to the Appendix of the revised manuscript. There, we discuss specific examples, such as redundancy arising from insufficient planning detail, as previously mentioned.
>
> [1] Dubois, Yann, et al. "Length-controlled alpacaeval: A simple way to debias automatic evaluators." arXiv preprint arXiv:2404.04475 (2024).
>
> [2] Li, Xuechen, et al. ‘AlpacaEval: An Automatic Evaluator of Instruction-Following Models’. GitHub Repository, GitHub, 5 2023, github.com/tatsu-lab/alpaca_eval.
>
> [3] Wang, Yizhong, et al. "Self-instruct: Aligning language models with self-generated instructions." Proceedings of the 61st annual meeting of the association for computational linguistics (volume 1: long papers). 2023.
>
> [4] Kopf, Andreas et al. “OpenAssistant Conversations - Democratizing Large Language Model Alignment.” ArXiv abs/2304.07327 (2023): n. Pag.
>
> [5] Chiang, Wei-Lin, et al. Vicuna: An Open-Source Chatbot Impressing GPT-4 with 90%* ChatGPT Quality. Mar. 2023, lmsys.org/blog/2023-03-30-vicuna/.
>
> [6] Geng, Xinyang, et al. Koala: A Dialogue Model for Academic Research. Apr. 2023, bair.berkeley.edu/blog/2023/04/03/koala/.
>
> [7] Bai, Yuntao, et al. "Training a helpful and harmless assistant with reinforcement learning from human feedback." arXiv preprint arXiv:2204.05862 (2022).
>
> [8] Jin, Tian, et al. "Learning to keep a promise: Scaling language model decoding parallelism with learned asynchronous decoding." Proceedings of the 42nd International Conference on Machine Learning (ICML). 2025.
>
> [9] Ning, Xuefei, et al. "Skeleton-of-Thought: Prompting LLMs for efficient parallel generation." Proceedings of the Twelfth International Conference on Learning Representations (ICLR). 2024.

---

### Official Review · Reviewer_p8N9 · 2025-11-03

**Soundness:** 2
**Presentation:** 2
**Contribution:** 3
**Rating:** 4
**Confidence:** 3

**Summary:**

* Problem: Autoregressive models produce high-quality text but are slow because they generate tokens sequentially, while diffusion models generate in parallel but often require many iterative steps to achieve similar quality.
* Solution: A new hybrid architecture, termed planned diffusion, that leverages (1) an autoregressive “planning” stage that decomposes the output into semantically independent spans, and (2) a parallel diffusion “execution” stage that denoises these spans simultaneously.
* Evaluation: Experiments and analysis including demonstrating that the method achieves a 1.84x speedup over autoregressive generation with a 6.8% drop in win rate on AlpacaEval.

**Strengths:**

1. Well-motivated: very relevant problem and one that addresses a key weakness in diffusion language models
2. Novelty: combining an autoregressive planning stage with a diffusion-based parallel generation stage within a single unified model.
3. Implementation: Proposes reasonable set of methods that includes a new control tag language, model training methodology, and inference algorithm that enable planned diffusion and navigation of a Pareto frontier between speed and performance.

**Weaknesses:**

1. Evaluation Scope: Evaluation is only on AlpacaEval and lacks any other benchmarks, tasks, or domains.
2. Baselines: There is only one baseline that is not the vanilla baselines of autoregressive models and diffusion LLMs.
3. Complexity: Quite a lot of complexity without full ablation to justify each design choice
4. Trade-off: A performance loss of 6.8% is still pretty substantial and it is not clear how much speed-up one could get with say a smaller model or speculative decoding.

**Questions:**

1. The improvement in speed comes at a cost which is performance. Performance is a much harder thing to raise, so it is hard to understand exactly how much is being sacrificed for the speedup. Is there a way to quantify the speedup with the performance kept constant or to show the performance with the same speed?
2. The following is mentioned: "To the best of our knowledge, this is the first text-only model that uses both discrete diffusion and autoregression." Does the emphasis of "text-only" mean that there are other models in different modalities that use both discrete diffusion and autoregression?
3. How does the model determine the optimal number and boundaries of spans in the autoregressive plan, and how sensitive are results to this segmentation?
4. Could you clarify how the computational cost of the diffusion stage scales with the number of spans and denoising steps?

---

> ### Author Response · Authors · 2025-11-22
> **Response to Reviewer p8N9 (1)**
>
> Dear Reviewer p8N9,
>
> We thank the reviewer for the constructive feedback and for recognizing the novelty and motivation of our proposed hybrid architecture. We also appreciate the acknowledgement of our implementation and the reasonable set of methods proposed. We have addressed your specific concerns and questions below.
>
> (1) Evaluation Scope
>
> Regarding the evaluation scope, we respectfully note that Length-Controlled AlpacaEval is a robust metric known to possess a high correlation with human preferences derived from Chatbot Arena [1]. Furthermore, AlpacaEval [2] is not a single task but a composite of five distinct datasets: self-instruct [3], open-assistant [4], vicuna [5], koala [6], and hh-rlhf [7]. This composition provides a holistic view of instruction-following capabilities across diverse settings; for instance, the self-instruct subset alone spans domains including email writing, social media, productivity tools, and programming (see Figure 3 of Wang et al. [3] for a complete taxonomy). We also present a per-dataset win rate breakdown in Appendix F, and observe that win rate is reasonably uniform across 5 datasets.
>
> (2) Baselines
>
> In addition to autoregressive (AR), Diffusion, and Fast-dLLM baselines, we evaluate two prior semantic-parallel decoding methods: Pasta-SFT [11] and Skeleton-of-Thought (SoT, [10]). Pasta-SFT implements semantic parallelism via control tags in a purely autoregressive model and, in our setting, we use its SFT-only variant (without RL) for a fair comparison to planned diffusion. SoT, by contrast, first samples a bullet-point skeleton autoregressively and then applies regex-based parsing to extract points for parallel expansion. Empirically, both lag behind planned diffusion: SoT is slightly faster than PD (3.10 s vs. 3.47 s) but suffers a notable drop in length-controlled win rate (40.7 % vs. 44.6 %), while in the absence of further RL training, Pasta-SFT fails to learn the control-tag semantics and yields substantially worse quality (26.1 %) at higher latency (5.74 s).
>
> (3) Complexity
>
> We appreciate the feedback regarding model complexity. We have conducted additional experiments alongside our existing ablations on training epochs, span lengths, and step ratios. We are happy to report interesting findings. First, we introduce Planned Diffusion Dense Attention (PD-DA): a key design decision was using block sparse masking to enforce independence between spans. In the revised manuscript, we ablate this by allowing dense attention among spans and find an increase in quality (see PD-DA in the updated Left Figure 3). Second, we ablate the control tags: our ablations demonstrate that removing the “topic” tag causes a significant quality drop, whereas removing the <sync> tag reduces quality but offers a large reduction in latency.
>
> (4) Tradeoffs
>
> Regarding the performance trade-off, as noted above, the dense attention variant (PD-DA) reduces the performance drop to only 0.8% compared to autoregressive baselines while maintaining a latency advantage. Regarding speculative decoding, we view it as an orthogonal approach that is difficult to compare directly for two reasons. First, speculative decoding requires training an auxiliary draft model, whereas planned diffusion does not. Second, speculative decoding is a fixed strategy that lacks the fine-tuned control over the speed-quality trade-off provided by our method, as demonstrated in our Quality Latency Sweep (Figure 4 right in the updated manuscript). For these reasons, while we acknowledge the approach, we focus on the comparison to diffusion and autoregression.

---

> ### Author Response · Authors · 2025-11-22
> **Response to Reviewer p8N9 (2)**
>
> **Questions**
>
> > (1) Is there a way to quantify the speedup with the performance kept constant or to show the performance with the same speed?
>
> Please refer to the Quality Latency Sweep in the updated manuscript. This figure highlights the full trade-off landscape by varying the number of diffusion denoising steps, allowing for a direct comparison of efficiency at comparable quality levels.
>
> > (2) Does the emphasis of "text-only" mean that there are other models in different modalities that use both discrete diffusion and autoregression?
>
> To the best of our knowledge, planned diffusion is the first hybrid autoregressive discrete diffusion model for any modality. We included the “text-only” qualifier to reduce the risk of overclaiming and to avoid confusion with respect to vision literature, where multimodal hybrids like HybridVLA [8] and Monoformer [9] utilize autoregression alongside continuous diffusion rather than discrete diffusion. We have updated the appendix of the manuscript to provide a more complete explanation of these distinctions.
>
> > (3) How does the model determine the optimal number and boundaries of spans in the autoregressive plan, and how sensitive are results to this segmentation?
>
> The number of spans is determined by our synthetic data pipeline, which labels the training data with control tags. As the number of spans increases, the model achieves greater parallelism; however, this may result in a drop in quality due to capturing fewer dependencies.
>
>
> > (4) Could you clarify how the computational cost of the diffusion stage scales with the number of spans and denoising steps?
>
> We have updated the manuscript to clarify the computational scaling. In a diffusion LLM, the inference cost is roughly the cost of attention multiplied by the number of iterations: $O(n^2 \cdot s)$, where $n$ is the sequence length and $s$ is the number of steps. Planned diffusion reduces the effective $s$ by a factor of $l/n$, where $l$ is the length of the longest span. This reduction is visualized in Figure 3, where the critical path $s$ is significantly lower than the baselines.
>
> [1] Dubois, Yann, et al. "Length-controlled alpacaeval: A simple way to debias automatic evaluators." arXiv preprint arXiv:2404.04475 (2024).
>
> [2] Li, Xuechen, et al. ‘AlpacaEval: An Automatic Evaluator of Instruction-Following Models’. GitHub Repository, GitHub, 5 2023, github.com/tatsu-lab/alpaca_eval.
>
> [3] Wang, Yizhong, et al. "Self-instruct: Aligning language models with self-generated instructions." Proceedings of the 61st annual meeting of the association for computational linguistics (volume 1: long papers). 2023.
>
> [4] Kopf, Andreas et al. “OpenAssistant Conversations - Democratizing Large Language Model Alignment.” ArXiv abs/2304.07327 (2023): n. Pag.
>
> [5] Chiang, Wei-Lin, et al. Vicuna: An Open-Source Chatbot Impressing GPT-4 with 90%* ChatGPT Quality. Mar. 2023, lmsys.org/blog/2023-03-30-vicuna/.
>
> [6] Geng, Xinyang, et al. Koala: A Dialogue Model for Academic Research. Apr. 2023, bair.berkeley.edu/blog/2023/04/03/koala/.
>
> [7] Bai, Yuntao, et al. "Training a helpful and harmless assistant with reinforcement learning from human feedback." arXiv preprint arXiv:2204.05862 (2022).
>
> [8] Liu, Jiaming, et al. "Hybridvla: Collaborative diffusion and autoregression in a unified vision-language-action model." arXiv preprint arXiv:2503.10631 (2025).
>
> [9] Zhao, Chuyang, et al. "Monoformer: One transformer for both diffusion and autoregression." arXiv preprint arXiv:2409.16280 (2024).
>
>
> [10] Ning, Xuefei, et al. "Skeleton-of-Thought: Prompting LLMs for efficient parallel generation." Proceedings of the Twelfth International Conference on Learning Representations (ICLR). 2024.
>
> [11] Jin, Tian, et al. "Learning to keep a promise: Scaling language model decoding parallelism with learned asynchronous decoding." Proceedings of the 42nd International Conference on Machine Learning (ICML). 2025.

---

> > ### Comment · Reviewer_p8N9 · 2025-11-28
> >
> > I appreciate the response and additional experiments, particularly the PD-DA variant, but I will maintain my score due to the continued reliance on a single benchmark (AlpacaEval) for evaluation.

---

### Author Response · Authors · 2025-11-22
**Summary of Changes**

We thank the reviewers for their valuable feedback. We have conducted additional experiments to strengthen the paper. We summarize our changes from most to least significant:

(1) Planned Diffusion Dense Attention (PD-DA)

To address concerns regarding the trade-off between speed and quality and to justify our design choices regarding span independence, we introduced a "Dense Attention" variant of our model. This variant relaxes the block-sparse masking constraint, allowing diffusion spans to cross-attend. As detailed in Figure 3 and Table 1, PD-DA significantly closes the quality gap, achieving a length-controlled win rate of 49.2% (compared to the autoregressive baseline of 50.0%) while maintaining a 1.27x speedup, thereby offering a highly competitive point on the Pareto frontier.


(2) Comparison to existing semantic parallelism


In response to requests for broader baselines, we have expanded our evaluation to include direct quantitative comparisons with other semantic parallelism methods. Specifically, we added results for Skeleton-of-Thought (SoT) and Pasta-SFT in Figure 3. These comparisons demonstrate that Planned Diffusion offers a superior balance of latency and quality compared to existing autoregressive parallel strategies.

(2) Quality Latency Sweep


To better quantify the trade-off between acceleration and performance, we performed a comprehensive hyperparameter sweep over the diffusion step ratio ($r$) and confidence threshold ($\tau$). The results, visualized in Figure 4 (Right), explicitly map the Pareto frontier, demonstrating that our method provides tunable control over inference speed and generation quality.

(3) Control Tag Ablation

To justify the complexity of our control tag language, we added an ablation study on the "topic" and "sync" tags (Figure 4, Left). These experiments confirm that semantic topic labeling is critical for maintaining generation quality. The synchronization tag offers higher quality at the cost of some speed.

(4) Qualitative Analysis

To address questions regarding failure modes and plan robustness, we added Appendix F: Qualitative Analysis. This section provides concrete examples of successful decomposition as well as failure cases (e.g., redundancy due to vague planning), offering deeper insight into the model's behavior and the importance of explicit semantic planning.

(5) General writing improvements

We have refined the manuscript to clarify our contributions and definitions. This includes clearer distinctions regarding "text-only" hybrid models to avoid confusion with multimodal works, improved explanations of computational scaling in the diffusion stage, and a breakdown of performance across the specific datasets within AlpacaEval (Appendix E) to address scope concerns.

---

### Comment · Area_Chair_LPvf · 2025-11-25
**Encourage discussions**

Hi all,

The authors have submitted their responses. Please take a moment to review them and see if they address your concerns.

Your thoughtful input is essential for a successful reviewing process and is greatly appreciated.

Many thanks,

Area Chair

---

### Meta-Review · Area_Chair_ECYv · 2025-12-17

**Summary:**

Most of concerns have been addressed by the authors. Therefore, AC would recommend Accept (poster).

To help the authors to prepare the camera-ready version, AC highlights that after the rebuttal, one common concern (lack of more benchmarks) remains, as stated by the reviewer. All reviewers had the same concern in the first round of review. The experiment is the bottleneck of the paper. AC would recommend the authors to conduct more solid and complete experiments in the camera-ready version.

**Reviewer Concerns:**

Reviewer concerns AC thinks were addressed by the rebuttal:

1. Trade-off: A performance loss of 6.8% is still pretty substantial and it is not clear how much speed-up one could get with say a smaller model or speculative decoding.

AC's comment: The additional experiments on Planned Diffusion Dense Attention close the gap.

2. Lack of baselines.

AC's comment: The concern has been resolved by additional experiments on Pasta-SFT [11] and Skeleton-of-Thought (SoT, [10]).

Reviewer concerns AC believes are still outstanding:

1. Continued reliance on a single benchmark (AlpacaEval) for evaluation.

AC's comment: This is a common concerns from multiple reviewers (Reviewer p8N9, Reviewer UEZx, Reviewer Zq7e, and Reviewer Yrr4). Reviewer p8N9 stated that he/she is not willing to change his/her score from being negative to positive. So the concern remains after the rebuttal.

**Reviewer Scores:**

The paper is on the borderline. Reviewer p8N9 stated that he/she will keep the score to be negative even after the rebuttal. Reviewer Yrr4 stated that he/she will keep the score the same (with a score of 6). So AC has confidence to believe that Reviewer Yrr4 is not excited about the rebuttal. Although two other reviewers do not express their opinions, AC believes with a high chance, they will keep the score unchanged after the rebuttal. Therefore, the scores will be most likely to be 4666 after the rebuttal. Given that three reviewers are positive, AC would follow the reviewers' recommendation and accept the paper as a poster.

---

### Decision · Program_Chairs · 2026-01-26

Accept (Poster)